# Usefulness of Aerobic Exercise for Home Blood Pressure Control in Patients with Diabetes: Randomized Crossover Trial

**DOI:** 10.3390/jcm11030650

**Published:** 2022-01-27

**Authors:** Keiko Iwai, Emi Ushigome, Kazufumi Okada, Isao Yokota, Saori Majima, Naoko Nakanishi, Yoshitaka Hashimoto, Hiroshi Okada, Takafumi Senmaru, Masahide Hamaguchi, Mai Asano, Masahiro Yamazaki, Michiaki Fukui

**Affiliations:** 1Department of Endocrinology and Metabolism, Graduate School of Medical Science, Kyoto Prefectural University of Medicine, Kyoto 602-8566, Japan; keiko816@koto.kpu-m.ac.jp (K.I.); saori-m@koto.kpu-m.ac.jp (S.M.); naoko-n@koto.kpu-m.ac.jp (N.N.); y-hashi@koto.kpu-m.ac.jp (Y.H.); semmarut@koto.kpu-m.ac.jp (T.S.); mhama@koto.kpu-m.ac.jp (M.H.); maias@koto.kpu-m.ac.jp (M.A.); masahiro@koto.kpu-m.ac.jp (M.Y.); michiaki@koto.kpu-m.ac.jp (M.F.); 2Department of Biostatistics, Graduate School of Medicine, Hokkaido University, Hokkaido 060-0808, Japan; okadak@pop.med.hokudai.ac.jp (K.O.); yokotai@pop.med.hokudai.ac.jp (I.Y.); 3Department of Endocrinology and Diabetology, Matsushita Memorial Hospital, Osaka 570-8540, Japan; conti@koto.kpu-m.ac.jp

**Keywords:** home blood pressure, randomized crossover trial, hypertension, type 2 diabetes, aerobic exercise

## Abstract

Hypertension usually coexists with diabetes mellitus and significantly increases the risk of macrovascular complications. Blood pressure measured at home, especially nocturnal blood pressure, is particularly important because it is more strongly associated with target organ damage than clinical blood pressure measurements. Regular moderate aerobic exercise has been shown to have anti-hypertensive effects. This study aimed to investigate the effects of aerobic exercise on home blood pressure in patients with diabetes. This randomized crossover trial was based on outpatient treatment at a university hospital. In this randomized crossover trial, 124 patients with type 2 diabetes were randomly assigned to two groups over 56 days: an exercise preceding group (exercise intervention for 28 days and then no exercise intervention for the following 28 days) and an exercise lagging group (no exercise intervention for 28 days and then exercise intervention for the following 28 days). The associations between the nocturnal blood pressure and exercise intervention were assessed accordingly. A decrease in blood pressure was observed in the morning and evening, at 2 a.m. and 3 a.m. after exercise intervention; however, there was no significant difference between groups. Moderate exercise was not effective in lowering nocturnal blood pressure in this study.

## 1. Introduction

### 1.1. Background

Patients with diabetes often have overlapping risk factors for arteriosclerosis, such as hypertension and dyslipidemia based on insulin resistance, which synergistically promote macrovascular complications [1]. The coexistence of hypertension and diabetes significantly increases the risk of macrovascular diseases, such as stroke and myocardial infarction [2,3]. Macrovascular complications are important factors in determining the long-term prognosis of patients with diabetes. To prevent its onset, early evaluation of arteriosclerosis and active intervention to control blood glucose, blood pressure, and lipid abnormalities are needed.

Intensive monitoring and treatment of hypertension and hyperglycemia are important for patients with diabetes. In particular, home blood pressure (HBP) is considered very important because it is more strongly associated with target organ damage, cardiovascular events, and total mortality than clinic blood pressure (BP) [4]. An acute bout of moderate aerobic exercise elicits a decrease in BP, known as post-exercise hypotension (PEH), which explains the clinical effects of aerobic exercise in hypertensive patients [5,6]. Furthermore, Kario et al. reported that the nighttime systolic BP (SBP), obtained using a home device, is a predictor of incident cardiovascular disease (CVD) events, independent of clinic or morning home SBP measurements [7]. Moderate exercise habits may be effective in controlling HBP, especially nocturnal SBP, in patients with diabetes.

### 1.2. Aims and Objectives

The primary aim of this study was to examine the usefulness of exercise therapy in controlling the nocturnal SBP from days 24 to 28 during the exercise period (EX period) compared with the non-exercise period (non-EX period) in patients with diabetes. The BP measured during the EX period was defined as the EX period BP and the BP measured during the non-EX period was defined as the non-EX period BP.

The second aim of this study was to compare the differences in the variation in nocturnal SBP, the mean SBP value, the variation in morning and evening SBP, hemoglobin A1c levels, serum lipid profile, renal function, liver function, and bioelectricity measured by InBody 720, between the EX period and the non-EX period. We hypothesize that, compared with the non-EX period, moderate exercise habits may be effective in controlling nocturnal SBP in patients with diabetes during the EX period, leading to a reduced risk of macrovascular complications.

## 2. Materials and Methods

### 2.1. Study Design

This study was a randomized crossover trial that aimed to evaluate the usefulness of exercise therapy in controlling HBP. Each participant was enrolled for a total of 56 days, from randomization to the final follow-up assessment. Overall, 124 patients with type 2 diabetes who received outpatient treatment at the hospital of the Kyoto Prefectural University of Medicine in Japan were randomly assigned to one of two groups: an exercise (EX) preceding group (exercise intervention for 28 days and then no exercise intervention followed by exercise intervention for the 28 days) and an EX lagging group (no exercise intervention for 28 days followed by exercise intervention for the following 28 days) [8]. All participants were asked to attend follow-up visits 1 and 2 months after enrollment. During enrollment, the patients were randomized to either the EX preceding group or the EX lagging group. Participants’ compliance during the EX and non-EX periods was confirmed using questionnaire records collected at the second follow-up visit at the end of the study.

After obtaining participants’ informed consent, a third party that was not involved in this research performed simple randomization using random numbers. All participants were randomized to one of two groups. Bliding was not possible for participants who requested to switch the EX period and the non-EX period for their convenience, as the participants allocated intervention during the trial would be revealed to explain the importance of the order of the intervention.

All procedures were approved by the Medical Research Ethics Committee of the Hospital of the Kyoto Prefectural University of Medicine (protocol reference number. R000040969) and were conducted in accordance with the principles of the Declaration of Helsinki. Informed consent was obtained from all patients.

### 2.2. Study Population

Target patients were recruited based on the following inclusion criteria: diagnosed with type 2 diabetes mellitus, aged between 20 and 90 years, and had written informed consent obtained before registration. The patients were also kept on the same medical treatment for at least 3 months before randomization to account for the effect of sodium-glucose transporter-2 (SGLT2) and glucagon-like peptide-1 receptor agonist (GLP1) on BP [9,10].

HBP measurements were performed in 124 patients with type 2 diabetes who regularly visited the diabetes outpatient clinic at the Hospital of the Kyoto Prefectural University of Medicine from June 2019 to March 2020. The diagnosis of type 2 diabetes was based on the Japan Diabetes Society’s Report of the Committee on the Classification and Diagnostic Criteria of Diabetes Mellitus [11].

### 2.3. Intervention

Instructions on activities permitted during the non-EX period and exercise intensity during the EX period were based on the metabolic equivalents of task [METs] table [12].

The exercise therapy included walking for 30 min twice a day, or 60 min once a day, with a goal of over 6000 steps per day, for more than 3 days per week. During the non-EX period, physical activity was restricted to activities of daily living (<4 METs) [8,12,13].

Patients assigned to the EX preceding group were instructed to exercise for at least 3 days per week from day 1 to day 28. After the follow-up visit 1 month after registration (Period 1), patients were instructed to perform the activities of daily living that were <4 METs) [8,12,13] from day 29 to day 56 until the follow-up visits 2 months after registration (Period 2).

Patients assigned to the EX lagging group were instructed to perform activities of daily living that were <4 METs from day 1 to day 28. After the follow-up visits 1 month after registration (Period 1), patients were instructed to exercise for at least 3 days per week from day 29 to day 56 until the follow-up visits 2 months after registration (Period 2).

### 2.4. BP Measurements

A validated and automatic device (HEM-7080IC; Omron Healthcare Co. Ltd., Kyoto, Japan) equipped with memory to store the date, time, and readings was loaned to each participant at baseline (first outpatient visit). The HBP was measured from day 24 to day 28 of each period (EX and non-EX period).

Patients were instructed to perform triplicate morning and evening BP measurements, with at least 1 min between recordings, for 5 consecutive days. Morning BP measurements were performed within 1 h of waking, before breakfast, or taking any medicines, with the patient seated and rested for at least 5 min. Evening BP measurements were obtained similarly just before bedtime. Patients were also instructed to go to bed with the cuff attached to the non-dominant arm, and the machine automatically measured the nocturnal BP at night, at 2:00, 3:00, and 4:00 a.m. In the original plan of the study, nocturnal BP was also measured at 5:00 a.m., as stated in the protocol, but it was excluded from the analysis because many participants were already awake at that time.

### 2.5. Definition

In this study, “dipper” is defined as BP decreasing by 10% to 20% during nighttime sleep. Compared to daytime, “extreme dipper” is defined as decreases of more than 20% during nighttime. “Riser” is defined as decreases less than 0% during nighttime sleep compared to daytime. “Non-dipper” is defined as “decreases by 0 to 10% during nighttime sleep compared to daytime” [14]. Non-dipping hypertension is associated with a higher degree of cardiovascular complications and more serious target organ damage than dipping hypertension [15].

Changes in BP patterns (riser, non-dipper, dipper, and extreme dipper) were compared before and after the exercise intervention. A change from riser, non-dipper, or extreme dipper to dipper was defined as an improvement of the pattern. A change from riser or non-dipper to extreme dipper and a change from riser to non-dipper were considered as semi-improvement of the pattern. A change from non-dipper, dipper, or extreme dipper to riser, a change from dipper or extreme dipper to non-dipper, and a change from dipper to extreme dipper were considered as deterioration of the pattern. 

### 2.6. Measurements and Outcomes

The primary endpoint was the mean nocturnal SBP at 2:00 a.m. during the EX period compared with the non-EX period. Since Kario et al. reported that the SBP at 2:00 a.m. was especially associated with CVD events [7], the primary endpoint of this study was revised from the protocol as the mean value of SBP at 2:00 a.m.

The secondary endpoints included the mean value of SBP at 3:00 a.m. and at 4:00 a.m., the variation in nocturnal BP, the mean value and variation of morning and evening SBP, and the following outcomes during the EX period compared with the non-EX period: hemoglobin A1c levels, blood glucose levels, serum lipid profiles, renal function, liver function, urinary albumin excretion levels, body weight, body mass index, and bioelectrical impedance analysis measured using InBody 720 [16].

Changes in diabetic and anti-hypertensive medications were recorded. Information on sex, age, the duration of diabetes, smoking status, alcohol consumption, the use of anti-hypertensive medication, and complications of diabetes were collected from each participant at baseline.

All outcome measures were assessed at baseline and at the first and second follow-up visits (Table 1). Details of the intervention and measurements are described in the protocol manuscript [17].

### 2.7. Statistical Analysis

Carlson et al. reported in a previous interventional study of hypertensive patients that exercise intervention reduced the systolic BP by approximately −7 mmHg, from 136 ± 12 mmHg to 129 ± 15 mmHg (*p* = 0.04) [18]. Based on this report, we set the standard deviation of systolic BP in each arm to 15 mmHg and the difference between the arms as 6 mmHg. To be conservative, the correlation between the arms was set to 0. With the one-sided significance level set at 0.025 and power set at 0.8, the required sample size was calculated as 101 participants under the paired *t*-test using SAS^®^ version 9.4 (SAS Institute Inc., Cary, NC, USA). Assuming a 10% dropout rate, we set the total sample size to 110.

The difference between the EX and non-EX period for all measures was used for analyses.

For the EX-preceding group, it was “Period 1-Period 2” and for the EX-lagging group, it was “Period 2-Period 1”.

Background characteristics were summarized by frequency and proportion (%) for categorical variables, and median and interquartile range for continuous variables, respectively. A paired *t*-test was used to analyze the primary and secondary endpoints before and after the exercise intervention to estimate the *p*-values and 95% confidence intervals (CIs), and to test the null hypothesis that the exercise lagging and exercise preceding groups had equivalent primary endpoints. JMP version 14.2.0 and SAS^®^ version 9.4 (SAS Institute Inc., Cary, NC, USA) was used for statistical analyses and a *p*-value of <0.025 (one-sided) was considered statistically significant.

## 3. Results

### 3.1. Participants

Of the 124 patients, we excluded those with progressive renal dysfunction (creatinine level ≥ 2.0 mg/dL) (*n* = 1) [19], those with inappropriate nocturnal BP recordings (*n* = 9), those who could not comply with the exercise instructions during the EX and/or non-EX period (*n* = 2), and those who declined to participate in the study (*n* = 15). Finally, 97 patients were included in the study population (53 men and 44 women; Figure 1).

### 3.2. Baseline Comparisons and Carry-Over Effect

The participants’ characteristics are presented in Table 2. There were 54 patients assigned to the EX preceding group and 43 patients assigned to the EX lagging group. There were no statistically significant differences at baseline between the groups for any outcome variables.

Assuming there was no carry-over effect, which is often a problem in crossover studies [20], both groups were combined and subjected to an anteroposterior *t*-test. The results of the two-sample *t*-test for the “EX period-non-EX period” of each group performed as a sensitivity analysis were not significant at the 5% level. Therefore, this assumption is valid (Table 3).

### 3.3. Outcome and Estimation

There was no significant difference between the primary endpoint, the mean SBP value at 2:00 a.m. during the EX and non-EX periods, and between the secondary endpoint and the mean SBP value at 3:00 a.m. and 4:00 a.m. However, there was a tendency for the SBP to decrease at 2:00 a.m. and 3:00 a.m., and to increase at 4:00 a.m. (Table 4, Figure 2).

None of the other secondary endpoints were significantly different, including the variations in nocturnal BP, the mean BP value, the variation in morning and evening BP between the non-EX period and EX periods, biochemical findings, body weight, body mass index, and bioelectrical impedance analysis (Table 4).

An examination of each nocturnal SBP pattern before and after the intervention revealed that the ratio of patients who improved to dippers was higher in those who were non-dippers (33.3%) and extreme dippers (33.3%) before the intervention, followed by risers. The ratio of patients with reduced nocturnal SBP was higher in those who were risers (50.0%) before the intervention. This was followed by non-dippers (33.3%) (Table 5).

## 4. Discussion

### 4.1. Principal Findings

This study did not demonstrate the effectiveness of moderate aerobic exercise on HBP in patients with type 2 diabetes.

Mean nocturnal SBP values were not significantly different; however, they tended to decrease at 2:00 a.m. and 3:00 a.m. after exercise intervention, while they tended to increase at 4:00 a.m.

Although not significantly different in our study, sustained BP elevation and BP variability (BPV) have been recognized as important predictors of CVD. Moreover, there seems to be a connection between high BPV and increased cardiovascular morbidity and mortality, even in patients with well-controlled BP [21,22]. These findings have been linked to alterations in vascular reactivity due to increased arterial stiffness and endothelial dysfunction [23].

By examining each nocturnal SBP pattern, we found that many patients who were non-dippers, extreme dippers, and risers during the non-EX period improved to dippers, which had a lower risk of CVD and target organ damage during the EX period. This increase in dipper status was thought to be due to the anti-hypertensive effect of exercise therapy. The mean nocturnal BP reduction was greater in the risers and the non-dippers who presented with higher BP than those who presented with near-normal nocturnal SBP. The results of this study are consistent with those of a previous report, wherein hypertensive middle-aged adults who were non-dippers responded to exercise treatment and had reduced nocturnal BP [15].

### 4.2. Interpretations

In patients with diabetes, exercise improves glycemic control [24], reduces BP [25], and improves cardiorespiratory fitness, which are factors associated with cardiovascular events and mortality [26].

Several intervention studies have reported improvements in BP with exercise therapy [27,28]. A decrease in peripheral vascular resistance and a reduction in cardiac output are associated with the occurrence of PEH [29]. Piepoli et al. reported that BP of healthy individuals performing maximal exercise decreases, probably because of a decrease in peripheral vascular resistance despite the increased renin activity, persistent sympathetic activity, and reduced vagal tone due to peripheral vasodilatation [30]. Several mechanisms have been proposed to explain the post-exercise decrease in peripheral resistance, including alterations in vascular responsiveness and sympathetic inhibition. As for the mechanism behind the anti-hypertensive effect, Kiyonaga et al. reported that an increase in nitric oxide and prostaglandin E due to exercise acts directly on vascular smooth muscle to relax blood vessels and reduce peripheral vascular resistance to lower the BP [31]. It has also been shown that a decrease in sympathetic nerve activity due to a decrease in blood catecholamines at rest leads to relaxation of peripheral blood vessels and a decrease in cardiac output [32].

Evidence has shown that exercise intensity influences the BP response of individuals with diabetes and hypertension. Eicher et al. reported that PEH was proportionate to the level of effort in the exercise sessions: the vigorous session (70–100% maximal oxygen uptake; VO_2max_) caused the greatest reductions in SBP and DBP compared with the control session [33]. Jones et al. also studied six normotensive individuals who performed aerobic exercises at 40% and 70% VO_2max_ in the morning and observed a mean arterial pressure reduction only during sleep after 70% VO_2max_ exercise, which suggests that the exercise intensity could have been responsible for this result [34]. Exercise performed at higher intensity mobilizes more motor units, resulting in greater metabolic and hemodynamic stress, as well as promoting a greater decrease in BP during the post-EX period.

Kario et al. has reported that nocturnal SBP is a predictor of incident CVD events, independent of clinic and morning home SBP measurements in the nationwide practice-based J-HOP Nocturnal BP Study [7]. Among the nocturnal BPs, the SBP at 2:00 and 3:00 a.m., especially the SBPs at 2:00 a.m., were shown to be strongly associated with cardiovascular events than the clinic BP. The results of our study are consistent with this report in that the SBP tends to decrease at 2:00 a.m. and 3:00 a.m. after exercise intervention, while it tends to increase at 4:00 a.m. Many patients exercise early in the morning and wake up early during exercise intervention period, which may lead to an increased BP at 4:00 a.m. due to the effects of insulin reverse-regulatory hormones such as cortisol and catecholamines [35].

This study has some limitations. First, the inclusion of patients with no history of macrovascular complications for exercise intervention compliance may have led to a selection bias. Approximately 10% of patients declined to participate in the study after entry owing to the time and effort required for exercise training and the follow-up visits. Second, participants’ compliance during the EX and non-EX periods was not monitored strictly. Only questionnaire records were collected at the end of the study to confirm the results. The use of smartwatches and pedometers would have been useful in making the intensity and duration of exercise uniform across participants and would also act as objective indicators. Third, we did not use devices such as actigraphs that provide movement-related information on sleep behavior [36]. Fourth, it was reported that the timing of exercise intervention was higher before sleep than in the morning [37]; however, the timing of exercise was not clear in our study. Finally, and most importantly, the intensity of exercise during the EX period may not have been sufficient. According to a study by Morais et al., a single aerobic exercise session resulted in BP reduction for 24 h in patients with type 2 diabetes, particularly while sleeping, and the magnitude of this reduction seemed to be dependent on the intensity at which the exercise was performed. After assessing functional performance, such as anaerobic threshold and VO_2max_ tests, it is important to prescribe the appropriate amount of exercise based on the patient’s aerobic and physical capacities.

## 5. Conclusions

The coexistence of hypertension and diabetes significantly increases the risk of macrovascular complications, such as myocardial infarction and cerebral infarction. Nocturnal SBP has been found to have a stronger relationship with target organ damage and total mortality than clinic BP.

Several studies have shown the effectiveness of aerobic exercise in reducing BP, especially during sleep, and that the degree of reduction appears to be dependent on exercise intensity. Therefore, although no significant difference was found in this study, it may not always be possible to say that aerobic exercise is not effective for nocturnal BP.

In the future, we would like to conduct further studies with higher exercise intensity by confirming exercise compliance more accurately using objective indicators, such as smartwatches and pedometers.

## Figures and Tables

**Figure 1 jcm-11-00650-f001:**
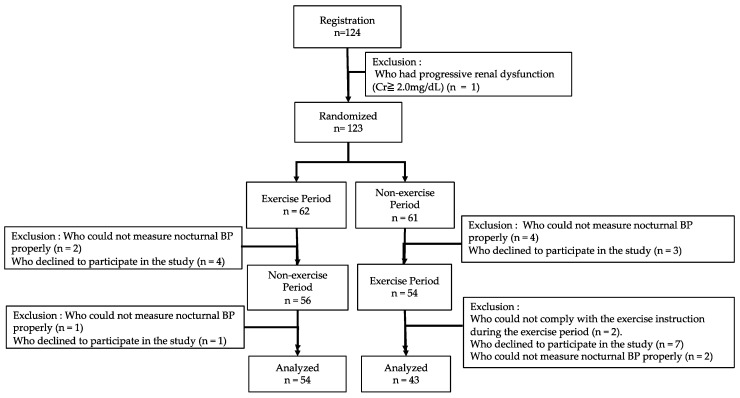
Flow diagram of the study.

**Figure 2 jcm-11-00650-f002:**
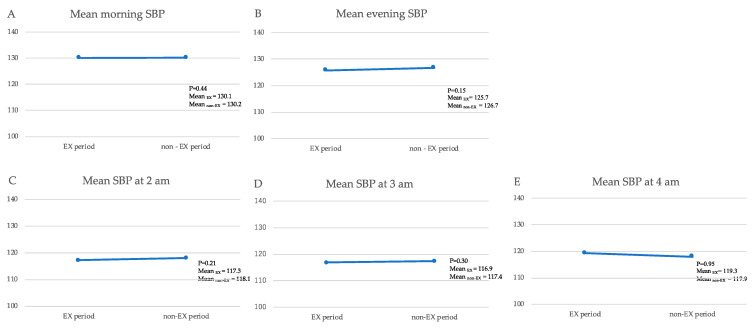
SBP at the non-EX period and the EX period (**A**) morning SBP (mmHg); (**B**) evening SBP (mmHg); (**C**) nocturnal SBP (mmHg) at 2:00 a.m.; (**D**) nocturnal SBP (mmHg) at 3:00 a.m.; (**E**) nocturnal SBP (mmHg) at 4:00 a.m.

**Table 1 jcm-11-00650-t001:** Overview of assessment schedule and measures.

			Study Follow-Up Visits	
TIMEPOINT	Enrolment	Allocation	1 Month	2 Months
ENROLMENT:				
Eligibility screen	X			
Informed consent	X			
Allocation		X		
Exercise preceding group Exercise period			X	
Exercise lagging group Exercise period				X
HBP measurement (days 24 to 28 each period)			X	X
ASSESSMENTS:				
Demographics	X			
InBody	X		X	X
Sleeping habits	X		X	X
Medication review	X		X	X
Body weight	X		X	X
Hemoglobin A1c	X		X	X
Blood glucose	X		X	X
Serum lipid profile	X		X	X
Renal function	X		X	X
Liver function	X		X	X
Urinary albumin excretion	X		X	X

HBP: home blood pressure.

**Table 2 jcm-11-00650-t002:** Clinical characteristics of patients.

Variables	Preceding	Lagging	Overall
	*n* = 54	*n* = 43	*n* = 97
Men	33 (61.1)	23 (53.5)	56 (57.7)
Women	21 (38.9)	20 (46.5)	41 (42.3)
Age (years)	69 (63–74)	69 (61–72)	69 (62.8–74)
Duration of diabetes mellitus (years)	9.5 (5–16)	14 (7–20)	12 (5.8–17)
Body mass index (kg/m^2^)	24.6 (22.4–26.8)	23.3 (21.6–26.3)	24.3 (22.0–26.6)
Hemoglobin A1c (%)	7.0 (6.5–7.8)	7.2 (6.7–7.8)	7.2 (6.6–7.8)
LDL cholesterol (mg/dL)	101.5 (87.5–121.3)	109.0 (86–122)	104.0 (87.5–122)
HDL cholesterol (mg/dL)	55.5 (46.8–69.0)	58.0 (50–68)	57.5 (48–68.3)
Triglycerides (mg/dL)	123.5 (93.0–207.8)	117 (88–159))	122.5 (92.5–181.8)
eGFR (mL/min/1.73 m^2^)	65.7 (57.7–82.1)	66.1 (55.5–77.1)	66.1 (57–77.3)
Smoking status			
current	6 (11.1)	2 (4.7)	8 (8.2)
past	18 (33.3)	15 (34.9)	33 (34.0)
never	30 (55.6)	26 (60.5)	56 (57.7)
Alcohol consumption status			
everyday	10 (18.5)	4 (9.3)	14 (14.4)
social	15 (27.8)	14 (32.6)	29 (29.9)
none	29 (53.7)	25 (58.1)	54 (55.7)
Diabetes complication			
Nephropathy (microalbuminuria)	18 (33.3)	15 (34.9)	33 (34.0)
Retinopathy	10 (18.5)	13 (30.2)	23 (23.7)
Neuropathy	14 (25.9)	8 (18.6)	22 (22.7)
Macrovascular complication	2 (3.8)	5 (11.6)	7 (7.2)
Use of antihypertensive medication	32 (59.3)	23 (53.5)	55 (57.0)
Use of SGLT-2 inhibitor	20 (37.0)	17 (40.5)	37 (38.0)

LDL: low-density lipoprotein; HDL: high-density lipoprotein; eGFR: estimated glomerular filtration rate; SGLT-2: sodium-glucose transporter-2. For categorical variables, *n* (%) was calculated. Continuous variables are presented as median (interquartile range).

**Table 3 jcm-11-00650-t003:** Sensitivity analysis: Confirmation that there is no carry-over effect (primary endpoint).

	Period 1	Period 2	EX Period–Non-EX Period
Preceding group	118.6 (114.4–122.8)	119.2 (115.3–123.1)	−0.6 (−3.6–2.5)
Lagging group	116.6 (113.0–120.2)	115.6 (111.8–119.4)	−1.0 (−3.0–0.9)
*p*-value *			0.8149

EX: exercise; * Student’s *t*-test *p*-value for “exercise-non-exercise”, comparison between groups.

**Table 4 jcm-11-00650-t004:** Comparison of primary endpoint and secondary endpoints analysis before and after exercise intervention (mean [95% CI]).

Variables	*n*	EX Period	Non-EX Period	EX Period-Non-EX Period	*p*-Value
Mean SBP at 2 a.m.	97	117.3 (114.4–120.1)	118.1 (115.4–120.1)	−0.8 (−2.7–1.1)	0.21
Mean SBP at 3 a.m.	97	116.9 (114.2–119.5)	117.4 (114.7–120.0)	−0.5 (−2.3–1.3)	0.30
Mean SBP at 4 a.m.	97	119.3 (116.5–122.1)	117.9 (114.9–120.8)	1.4 (−0.3–3.2)	0.95
Mean morning SBP	87	130.1 (126.9–133.2)	130.2 (127.1–133.3)	−0.1 (−2.0–1.7)	0.44
Mean evening SBP	94	125.7 (122.3–129.1)	126.7 (123.4–130.0)	−1.0 (−2.8–0.9)	0.15
BP variation at 2 a.m.	97	10.26 (9.25–11.28)	10.28 (9.31–11.26)	−0.02 (−1.37–1.32)	0.49
BP variation at 3 a.m.	97	9.89 (8.80–10.99)	10.59 (9.48–11.71)	−0.70 (−2.15–0.75)	0.17
BP variation at 4 a.m.	97	9.50 (8.41–10.58)	9.67 (8.51–10.83)	−0.17 (−1.50–116)	0.40
BP variation at morning	87	3.53 (2.99–4.06)	3.59 (3.07–4.12)	−0.07 (−0.64–0.50)	0.40
BP variation at evening	94	3.46 (2.97–3.96)	3.57 (3.09–4.05)	−0.10 (−0.55–0.34)	0.32
Hemoglobin A1c	97	7.18 (7.02–7.34)	7.24 (7.07–7.41)	−0.66 (−0.12–0.00)	0.03
Blood glucose	97	145.6 (138.1–153.1)	153.2 (143.2–163.1)	−7.6 (−17.5–2.4)	0.07
LDL	97	107.4 (101.7–113.1)	109.4 (103.5–115.4)	−2.0 (−5.5–1.5)	0.12
Triglycerides	97	149.2 (129.9–168.6)	147.5 (128.8–166.2)	1.7 (−15.1–18.6)	0.58
Creatinine	96	0.82 (0.77–0.87)	0.82 (0.77–0.87)	0.00 (−0.01–0.02)	0.74
AST	97	27.2 (23.9–30.6)	25.2 (21.9–28.5)	2.0 (0.3–3.8)	0.99
ALT	97	27.3 (22.6–32.0)	27.7 (23.0–32.4)	−0.4 (−2.1–1.3)	0.32
Urinary albumin excretion	81	102.0 (46.3–157.8)	125.7 (23.3–228.1)	−23.7 (−89.7–42.3)	0.24
Body weight	97	64.8 (62.0–67.5)	65.0 (62.2–67.7)	−0.2 (−0.4–0.0)	0.02
Body mass index	97	24.5 (23.7–25.3)	24.5 (23.7–25.4)	0.0 (−0.1–0.0)	0.16
SMI	97	6.95 (6.69–7.22)	7.08 (6.81–7.35)	−0.13 (−0.35–0.10)	0.13

EX: exercise; SBP: systolic blood pressure; LDL: low-density lipoprotein cholesterol; AST: aspartate aminotransferase; ALT: alanine aminotransferase; SMI: skeletal muscle index; CI: confidence interval. Continuous variables are presented as median (interquartile range).

**Table 5 jcm-11-00650-t005:** The number of each nocturnal SBP pattern at the non-EX period and at the EX period and the ratio of people with reduced and improved the nocturnal BP.

	EX Period BP Pattern (Number)				Percentage of People with Decreased Nocturnal BP (%)	Percentage of People Who Improved to Dippers (%)
Non-EX Period BP Pattern (Number)	Non-Dipper	Dipper	Extreme Dipper	Riser		
Non-dipper (*n* = 24)	14	8	0	2	33.3	33.3
Dipper (*n* = 41)	9	26	4	2	9.8	N/A
Extreme dipper (*n* = 18)	1	6	12	0	0	33.3
Riser (*n* = 14)	6	1	0	6	50	7.1

EX: exercise, N/A: not applicable.

## Data Availability

Not applicable.

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
