# Peer review of "Usefulness of Aerobic Exercise for Home Blood Pressure Control in Patients with Diabetes: Randomized Crossover Trial"

_jcm, 2022, doi:10.3390/jcm11030650_

Round 1

Reviewer 1 Report

Thank you for addressing my earlier comments. The study design and analyses are now sufficiently clear to me, as are your conclusions. I have no further comments.

Author Response

We greatly appreciate your review of our manuscript and helpful suggestions. I am grateful for the opportunity to re-analyze and revise this research based on your suggestions. Thank you.

Reviewer 2 Report

Thank you for the opportunity to review this paper.

The statistical analysis section should be improved - the methods applied to compute descriptive statistics should be included.

Also, sample size determination should be included.

The manuscript was difficult to follow since it was a pdf file with track-changes.

Author Response

We greatly appreciate your review of our manuscript and helpful suggestions. We have revised the Statistical analysis section. The determination of the sample size described in the protocol paper has been added to the manuscript (P5L679-P5L686).

The methods applied to compute descriptive statistics has been added to the manuscript (P5L690-P5L692). Along with this revision, the explanation of Table 4 has been added as follows(P7L805).

I am sorry that the manuscript was difficult to understand due to a major revision. I will attach the WORD version of the revised manuscript.

In addition, we requested Editage to proofread the English text again and revised the manuscript.

This manuscript is a resubmission of an earlier submission. The following is a list of the peer review reports and author responses from that submission.

Round 1

Reviewer 1 Report

The manuscript describes the analysis of a randomised trial to investigate the effect of aerobic exercise on home blood pressure in type 2 diabetes patients. The trial consists of a two-period two-treatment cross-over design with about 100 patients randomised to a ‘no exercise’ followed by ‘exercise’  respectively a ‘exercise’ followed by ‘no exercise’ group with 28 days per period. Home blood pressure measurements are available at study entry and at the end of each period (averaged over several days).

The manuscript addresses a relevant research question as regular exercise is supposed to provide better blood pressure control for diabetes patients, which in turn can reduce longer-term complications. The study design, study population, and measurements are reasonably well described. However,  I find several severe problems with the manuscript that overall result, I will not recommend the manuscript in its current form.

1) There are large inconsistencies regarding the study question and relevant background between Abstract, Introduction, Methods, Results, and Discussion/Conclusion. For example, the introduction argues that arteriosclerosis is a main complication of interest, but this is never mentioned in the abstract and is only a minor part of the discussion. The relation and distinction between the different types of blood pressure measurements (home / office / clinic, systolic, nighttime) are never properly explained or their relevance discussed.

2) Particularly Section 4.2, the interpretation of the study’s findings, provides a list of BP-related facts and studies, but these are not related to each other or to the results and rationale of this manuscript.

3) I do not see the rationale for using a cross-over design for this study. What relevant information could one extract by comparing the two orders of ‘exercise’ and ‘no exercise’? 

4) Provided a cross-over design is reasonable, the subsequent analysis should take account of this design. However, the authors state to use a paired t-test for the primary analysis. I agree that within each treatment group a paired t-test would consider within-patient contrasts between the two periods, but how are these compared between the two treatment groups? What is the rationale for this comparison? 

5) The primary study endpoint is a comparison between the nocturnal BP in the exercise and the no exercise periods. I assume this ‘comparison’ is a difference of averages? Period-effects and period-treatment interactions  from the cross-over design would alter the interpretation of this difference if unaccounted for.

6) What seems to be the primary analysis are independent tests and estimates for BP at different points during the day. Figures 2-6 provide comparisons of ‘Initial’ to ‘Final’ BP, but it is not explained what these mean. In lines 140-141, the authors discuss ‘before and after intervention’, but what exactly is ‘before intervention’ for the ‘exercise lagging group’ for example? From my understanding, these figures present averages over all measurements and patients during the ‘exercise’ versus ‘no exercise’ periods, which would make the cross-over design irrelevant. 

7) In all cases (Figs 2-6), the effect size (average difference in BP) is between 0 to 4 mmHg, a relative change of less than 4%. Given the standard deviation in each case is already about 15-20 mmHg, I do not see how any valid statement about increases or decreases in BP can be made from these data. I certainly do not agree that the claimed differences are principal findings. Note that I do not object to presenting results that do not reach statistical significance, but to presenting results with minute and clinically irrelevant effect sizes as relevant findings.

8) I do not understand the multivariate logistic regression. While the authors present a list of independent covariates for their model, they do not explain the dependent response variable. For a logistic regression, the response is a 0/1, yes/no, success/failure-type variable, but what is it in this case specifically? In addition, the authors present no results for this model. I would like to see at least the parameter estimates and relevant contrasts and their uncertainties or p-values. Instead, the authors provide the result for a single covariate (line 151) with a p-value failing their own significance standard of 2.5%, which is then taken up again as a principal finding. The variable is ‘macrovascular complications’ for which no data is provided in Table 1. 

9) The study population is very heterogeneous with large ranges of body weight and age. More problematic seems to be the large percentage of patients with blood pressure medication (Table 1). Without adjusting for this medication, it seems difficult to compare patients across treatment groups properly. In addition, no adjustments are made for any of these variables in the primary analyses; this does not necessarily impede the validity of the analyses, but sacrifices power and precision, a major problem in this study.

10) The subgroup analyses lack justification and proper presentation. In particular, what are the remaining number of patients for each subgroup? Would one not expect large correlations between these groups, such as users of hypertensive drugs also having larger BMI or more likely to have had macrovascular complications? As presented, I do not see what additional information can be gained by these analyses.

11) I do not understand the analysis summarised in Table 2: How are the different responder groups defined (i.e., what is a ‘Dipper’)? No relevant source is cited or explanation given. Table 2 also provides information for changes in BP patterns, which would lend itself for further analysis on the significance of, e.g., the number of ‘Risers’ becoming ‘Dippers’, but no such analysis is given. Also here, I am unsure what is meant by ‘Initial BP’ and ‘Final BP’. 

12) The ‘no exercise’ periods allow for <4 MET for daily life, while the ‘exercise’ periods requires a total of at least 60min walking at least 3 times per week. However, this amount of exercise corresponds to about 2-3 MET, roughly the same as ordinary household chores such as vacuuming. This seems to provide very little difference in overall daily physical activity between the ‘exercise’ and ’no exercise’ periods, which would partially explain why essentially no differences are found between the two periods.

13) In principle, the measurements taken from the same patient provide a nice time-course of BP even over several days, but this information is never used. Instead, the authors present averages of BP over both treatment groups and all patients, which negates the use of multiple measurements per patients and the cross-over design. I suspect that much more information could be extracted from this dataset by appropriate statistical analysis using techniques to account for longitudinal measurements and the cross-over design, which would likely reduce the large uncertainty of comparisons considerably. The same holds for the fact that BP measurements are taken on several consecutive days at the end of each period for each patient. This would allow estimation of within- and between-patient variability, the prevalence of ‘BP patterns’ (see 11) and the frequency with which individual patients show different ‘BP patterns’.

Overall, I believe that this study has the potential to show interesting and relevant results. However, the analyses provided in the manuscript are severely flawed, lack most of the relevant information to understand and gauge the results, and are largely inconsistent with the study design. I would recommend that the authors seek the assistance of a statistician to analyse this trial properly. In its current form, I reject the publication due to the listed inadequacies.

Reviewer 2 Report

  1. Please, correct punctuation throughout the text including abstract

(for example:  “In addition; regular moderate aerobic exercise has antihypertensive effects.” Probably this is not a place for a semicolon.

  1. On describing Figures I would recommend to avoid the expression “of those who do not use…” and to exchange it for more formal manner such as “of patients with…etc.”
  2. In Conclusions :

“The coexistence of hypertension and diabetes significantly increases the risk of macrovascular diseases, such as myocardial infarction and cerebral infarction. The nocturnal BP has been found to have a stronger relationship with target organ damage and total mortality than the clinic BP. “

…. this sentence is an introduction to final Conclusion I assume.

However, the next one is illogical for the use of comma instead of semicolon before the word “demonstrating”. Please, re-arrange this part.

The coexistence of hypertension and diabetes mellitus increases the risk of macrovascular complications, demonstrating that moderate exercise therapy is effective for controlling the nocturnal BP in patients with diabetes would have great clinical significance.”

  1. The rest of the article is understandable and even though the result of the study was as it was, it still presents the interesting ideas.